# Changes in Barriers That Cause Unmet Healthcare Needs in the Life Cycle of Adulthood and Their Policy Implications: A Need-Selection Model Analysis of the Korea Health Panel Survey Data

**DOI:** 10.3390/healthcare10112243

**Published:** 2022-11-09

**Authors:** Woojin Chung

**Affiliations:** Department of Health Policy and Management, Graduate School of Public Health, Yonsei University, Seoul 03722, Korea; wchung@yuhs.ac; Tel.: +82-2-2228-1522

**Keywords:** unmet healthcare needs, non-financial barriers, time constraints, lack of caring and support, universal health coverage, longitudinal study, Korea Health Panel survey, panel multinomial probit model with sample selection, South Korea

## Abstract

Using 68,930 observations selected from 16,535 adults in the Korea Health Panel survey (2014–2018), this study explored healthcare barriers that prevent people from meeting their healthcare needs most severely during adulthood, and the characteristics that are highly associated with the barrier. This study derived two outcome variables: a dichotomous outcome variable on whether an individual has experienced healthcare needs, and a quadchotomous outcome variable on how an individual’s healthcare needs ended. An analysis was conducted using a multivariable panel multinomial probit model with sample selection. The results showed that the main cause of unmet healthcare needs was not financial difficulties but non-financial barriers, which were time constraints up to a certain age and the lack of caring and support after that age. People with functional limitations were at a high risk of experiencing unmet healthcare needs due to a lack of caring and support. To reduce unmet healthcare needs in South Korea, the government should focus on lowering non-financial barriers to healthcare, including time constraints and lack of caring and support. It seems urgent to strengthen the foundation of “primary care”, which is exceptionally scarce now, and to expand it to “community-based integrated care” and “people-centered care”.

## 1. Introduction

The goal of the health system, which most countries aim for, is to ensure that the entire population has access to the necessary healthcare, regardless of their social and economic conditions. Thus, many countries strive to identify and minimize barriers that cause unmet healthcare needs because they lead their citizens to not properly receive the necessary healthcare on time [1,2]. Unmet healthcare needs have been defined as occurring when individuals think that they need healthcare for particular health conditions but do not actually receive it. In many previous studies, the information has been derived from survey respondents’ answers to whether they have felt that they needed healthcare but did not receive it, and the reason they did not, in the past 12 months [3,4,5,6,7,8,9,10,11,12,13].

Unmet healthcare needs have been reported that their socioeconomic disparity is substantially large in most countries. Compared with high-income people, low-income people have higher unmet healthcare needs [3]. Therefore, many studies have highlighted the harmful effects of financial barriers on healthcare utilization [14,15]. As a result, many countries have focused on lowering financial barriers to healthcare [3,16,17]. These efforts include achieving universal health insurance and expanding and deepening financial coverage for various healthcare services through public and private health insurance.

However, recent studies suggest that academics and governments should pay attention to non-financial barriers to healthcare [4,5,7,9,18,19]. These studies argue that access to healthcare is not achieved if non-financial barriers are prevalent, even in countries with universal health insurance systems where everyone can receive free healthcare.

In the past, several studies have contributed significantly to addressing various barriers to healthcare [10,11,12,20]; however, problems with classifying barriers to healthcare in these studies have created difficulties in accurately understanding the results and making them fruitful with effective policies to reduce unmet healthcare needs. For example, because the barriers were classified, bundled, and analyzed according to specific dimensions of the group, such as affordability and accessibility, it was difficult to explore how significantly a specific barrier affects individuals’ access to healthcare than others and reflect the results in policy development. It was difficult to identify characteristics that were significantly related to a specific barrier to healthcare.

Meanwhile, without depending on the classification problem mentioned previously, a recent study examined the association between social capital and unmet healthcare needs in Europe using a bivariate sample selection model, where unmet healthcare needs are conditional on healthcare needs [13]. This study was novel and interesting in combining a sample selection model with social capital theory. However, this study has some limitations: (1) it did not use longitudinal data, and (2) it did not explore which of the specific barriers is likely to increase people’s unmet healthcare needs.

Another recent study examined national longitudinal data using a panel-data mixed logit choice model [9]. This study emphasized the importance of non-financial barriers to healthcare and provided many policy suggestions. However, its limitations are (1) it did not separate people based on whether they needed healthcare, (2) it did not divide people who needed healthcare based on whether their healthcare needs were met, (3) it did not divide non-financial barriers that led to unmet healthcare needs precisely, and (4) it did not use survey weights for methodological reasons.

This study tried to overcome the limitations of previous studies by exploring specific barriers that cause people not to meet their healthcare needs during adulthood and people’s characteristics that are highly associated with the specific healthcare barrier. To achieve this, this study used (1) a nationally representative panel survey dataset; (2) analyzed the study sample using a multivariable, panel multinomial probit model with sample selection, where both met and unmet healthcare needs were conditional on the existence of healthcare needs; and (3) estimated the probability that an individual would experience healthcare needs and the probability that an individual would experience a specific type of barrier that led to unmet healthcare needs.

This study’s findings are expected to contribute to developing and testing of new theories regarding barriers to healthcare access. In addition, these findings will guide policymakers in establishing target policies that help people meet their healthcare needs. This study examines the case of South Korea. Considering that the country has a universal health insurance system and is facing rapid population aging with a rapid increase in national health expenditures, this study could provide insight into countries worldwide with healthcare system environments similar to South Korea.

## 2. Brief Overview of South Korea’s Healthcare System

South Korea (hereafter, Korea) facilitates access to healthcare services for the entire population through two types of national health security programs, namely, the National Health Insurance (NHI), which is a social health insurance program, and Medical Assistance (MCA), which is a public in-kind assistance program for the poor. The MCA program covers approximately 3% of the population [9,21].

NHI is administered by the National Health Insurance Corporation (NHIS), a single public insurer, and is strongly guided and supervised by the Ministry of Health and Welfare of Korea. NHI is mainly funded through contributions to the payroll income of employees and their property and the estimated income of self-employed and agricultural people. Because the NHI’s governance structure is centralized, the contribution schedule and benefits packages are the same nationwide. Healthcare delivery relies heavily on private providers, and physicians and hospitals are largely reimbursed using the fee-for-service payment method. The framework for traditional primary care is exceptionally weak and most physicians are specialists. When people are sick, their first contact with a physician is mostly with specialists, whose healthcare services are competitively provided at all levels of health facilities.

Referring to a recent report by the Organization for Economic Cooperation and Development (OECD) [3], the current status of Korea’s healthcare system is summarized as follows: Health expenditure as a percentage of the gross domestic product was 8.4% in Korea in 2020, approaching its average value in 38 OECD member countries (8.8% in 2019). The annual growth rate in health expenditure per capita between 2015 and 2019 was 7.8% in Korea, the highest in 38 OECD member countries, except for Latvia (8.0%), whereas its average value in those countries was 2.7%.

As for access to healthcare, whereas the number of annual doctor consultations reported per capita was, on average, 6.8 in 34 OECD member countries in 2019, its value was 17.2 in Korea in 2019, the highest among those countries. Similarly, the average length of hospital stay was 18.0 days in Korea in 2019, the highest among 38 OECD member countries, compared to 7.6 days, its average value in those countries. On average, the number of beds per 1000 people was 4.4 in 38 OECD member countries in 2019; however, Korea’s number was 12.4, which was the second-highest value after Japan’s 12.8.

Regarding the health workforce, the number of practicing doctors per 1000 people was 2.5 in Korea in 2019, far behind 3.6, its average value in 38 OECD member countries. In particular, the share of general practitioners of all doctors was 6% in Korea in 2019, which is the lowest among 32 OECD member countries (the average value is 23%). Meanwhile, the number of practicing nurses per 1000 people was 7.9 in Korea in 2019, which was lower than 8.8, its average value in 38 OECD member countries.

As for health status and aging, life expectancy at birth was 83.3 years in Korea in 2019, which is higher than 81.0 years, the average value of 38 OECD member countries. The percentage of the population aged 65 years and above was 14.9% in Korea in 2019, which was lower than 17.3%, the average value in 38 OECD member countries.

In addition, for the proportion of adults rating their health as bad or very bad, its average value in 36 OECD member countries was 8.5% in 2019, but Korea showed 15.2%, its highest value, except for Latvia (15.4%). Likewise, deaths by suicide per 100,000 people (age-standardized) were 24.6 in Korea in 2019, which was the highest among 38 OECD member countries (their average value was 11.0 in 2019). Regarding the volumes of second-line antibiotics in terms of DDD (defined daily dose) per 1000 people per day, it was 9.4 in Korea in 2019, which is the highest except for Greece (10.6) (its average value in 30 OECD member countries is 3.3).

## 3. Materials and Methods

### 3.1. Data Source and Study Sample

This study results from extensive research to identify barriers that cause unmet healthcare needs and derive policy implications necessary to reduce them. Therefore, although the statistical methods in this study are more sophisticated than those in the previous one [9], it is necessary to report in advance that this study has some similarities with the previous study regarding research data and methods. Research data and methods similar to those in the previous study may be omitted, but for the convenience of readers, detailed explanations will be provided as follows.

This study analyzed data collected from the Korea Health Panel (KHP) survey (version 1.7), a nationally representative, non-institutionalized civilian population survey. The KHP survey was conducted by the NHIS and the Korea Institute of Health and Social Affairs, a state-run research institute, under the direction and supervision of the Ministry of Health and Welfare.

In the KHP survey, households were selected using a two-stage cluster probability sample of population census data provided by the National Statistical Office. The survey included data from all eligible household members related to individual healthcare use, health expenditures, social demographics, lifestyle, and health-related characteristics. Data were collected using computer-aided personal interview techniques once a year during notified weekdays. Detailed information regarding the survey is available on the KHP website (https://www.khp.re.kr:444/eng/main.do, accessed on 20 May 2020).

Although the survey began in 2008, this study used data from 2014 to 2018 for two reasons. One of the reasons for this is that in 2014, there was a change in the method of reporting datasets on chronic disease-related information. Second, since the COVID-19 pandemic began in 2019, data in and after 2019 related to healthcare use seem likely to be temporarily affected by the pandemic. This study involved 72,867 observations for individuals aged 19 years or older; however, it excluded observations without information on healthcare needs and unmet healthcare needs (3928) and those without explanatory variables (9). Therefore, the final study sample comprised an unbalanced panel sample of 68,930 observations (94.6%) of 16,535 individuals, with an average of 4.2 observations per individual (standard deviation = 1.4, range = 1 to 5). In the study sample, 11,582 people contributed five times, 868 people contributed four times, 1032 people contributed three times, 1414 people contributed twice, and 1644 people contributed once.

### 3.2. Measurements

#### 3.2.1. Outcome Variables

First, we sought to construct two outcome variables. One outcome variable was whether an individual had experienced healthcare needs or not. The other outcome variable is whether the healthcare needs of the individual who has experienced healthcare needs have been met, and if not, why had they not been met.

By scrutinizing the KHP survey questionnaire, this study identified two questions that could help derive the two outcome variables. The first question was “Have you ever experienced not receiving the necessary medical treatment or examination (excluding dental care) in the past year (12 months)?” The answers that each individual could choose were as follows (multiple answers were not allowed): (1) Yes, I have experienced it at least once; (2) No, I have not experienced it; and (3) No medical treatment or examination of any kind was needed.

For the individual who answered, “(1) Yes, I have experienced it at least once”, the question was followed by “Which of the following is the main reason for not receiving the necessary medical treatment or examination?” For this question, the answers that each individual could choose were as follows (multiple answers were not allowed): (1) financial reasons (medical expenses); (2) health facilities are far away; (3) functional limitations or poor health make it difficult to visit a health facility; (4) no one cares for children; (5) symptoms were not severe; (6) I had no information on where to go, (7) no time to visit a health facility, (8) I could not make a reservation at a proper time, (9) I have no regular doctor, and (10) other reasons.

This study first focused on answering the first question to construct the two outcome variables mentioned above. This study divided all participants into two groups according to their answers. Those who answered “(1) Yes, I have experienced it at least once” or “(2) No, I have not experienced it” were classified into the “needs” group. In addition, those who answered “(3) No medical treatment or examination of any kind was needed” were classified as the “non-needs” group. Subsequently, a dichotomous outcome variable was constructed where its value is “1” if an individual belongs to the “needs” group and its value is “0” if the individual belongs to the “non-needs” group.

This study focused on individuals who had experienced healthcare needs (the “needs” group) to construct the second outcome variable. The individuals who answered “(2) No, I have not experienced it” to the first question are those who have experienced healthcare needs and have received the necessary healthcare services. Therefore, they are classified as individuals whose healthcare needs have been met, that is, the “met needs” group.

However, among the “needs” group, individuals who answered (1) “Yes, I have experienced it at least once” to the first question were those who have experienced healthcare needs but failed to meet their own healthcare needs for some reason. Therefore, they were classified as individuals who failed to meet their healthcare needs, that is, the “unmet needs” group. The reason why individuals who belong to the “unmet needs” group failed to meet their healthcare needs can be understood by investigating their answer to the second question. Therefore, according to the answers of the individuals in the “unmet needs” group to the second question and considering their characteristics and distribution, this study categorized them into three groups, focusing on their healthcare barriers: (1) financial difficulty, (2) time constraint, and (3) lack of caring and support.

First, individuals who chose as their answer to the second question the “(1) financial reasons (medical expenses)” option were categorized into the “financial difficulty” group. Second, individuals who answered the “(7) no time to visit a health facility” option were categorized into the “time constraint” group. Third, those who chose any one of the listed reasons apart from the two reasons regarding the “financial difficulty” and “time constraint” groups were categorized into the “lack of caring and support” group because if those people had received tailored caring and support, they would have been able to meet their healthcare needs.

Consequently, this study successfully constructed two necessary outcome variables. One is a dichotomous outcome variable (the “needs” and “non-needs” groups) regarding whether an individual had experienced healthcare needs. The other is a quadchotomous outcome variable (the “met needs”, “financial difficulty”, “time constraint”, and “lack of caring and support” groups) regarding how the healthcare needs of the individual who had experienced healthcare needs (the “needs” group) ended up.

#### 3.2.2. Explanatory Variables

This study included various explanatory variables in its analysis, such as sociodemographic characteristics and physical and health conditions. Owing to the reasons mentioned in Section 3.1, these explanatory variables are the same as in previous studies [9]. Although the reader could be guided to refer to the previous paper without including this part, this study presents the explanatory variables below for the reader’s convenience.

Regarding sociodemographic characteristics, this study included gender (man or woman); age; marital status (married or unmarried; residential area (Seoul metropolitan area or areas outside of the Seoul metropolitan area); the highest level of formal education completed (lower than college, college, or higher); occupation (no job, blue-collar job, or white-collar job); household income (lowest quintile, medium, or highest quintile); national health security programs status (the NHI or the MCA); and private health insurance status (yes or no).

Those who were unmarried included those never married, separated, widowed, or divorced. The Seoul metropolitan area includes the Seoul, Incheon, and Gyeonggi provinces. No job included the unemployed or individuals outside of economic activity, such as housekeepers, students, and retired individuals. Household income was adjusted for household size using the square root equivalence scale for each wave, and the medium included the three middle quintiles) [22]. The private health insurance status indicates whether an individual is a beneficiary of at least one private health insurance plan.

In this study, the physical and health conditions considered were (1) functional limitation (yes or no), (2) currently smoking (yes or no), (3) alcohol consumption (yes or no), (4) an active routine of physical exercise activity (yes or no), (5) obesity (yes or no), (6) poor self-assessed health (yes or no), and (7) chronic disease (yes or no).

To explain these variables in detail, a functional limitation was determined based on an individual’s answer to the question: “Is your daily living routine (conducting work, housekeeping, study, and social, leisure, or familiar activities) limited due to a disease or an injury?” The active routine of physical exercise activity was defined based on an individual’s answer when assessing their engagement in any three kinds of physical exercise (walking, medium-level, or high-level exercise) for 30 min or longer, at least three times a week. Obesity was defined as an individual’s body mass index of at least 25 kg/m^2^, which complies with the recommendation in the Asia-Pacific criteria concerning obesity status provided by the World Health Organization Western Pacific Region, based on an individual’s answer to questions on height and weight [23]. Poor self-assessed health involved an individual’s self-rating of their general health as “poor” or “very poor” among the options of “excellent, very good, fair, poor, or very poor.” Chronic disease was determined based on self-reported answers regarding whether a physician diagnosed an individual as suffering from any chronic disease at the time of the survey.

### 3.3. Statistical Analysis

When investigating which group an individual belongs to among the “met needs”, “financial difficulty”, “time constraint”, and “lack of caring and support” groups, it is important to note that these can be observed only among individuals who have experienced healthcare needs. Otherwise, the results of the analysis might suffer from sample selection bias [24,25]. This study used a multivariable panel multinomial probit model with sample selection to correct for potential sample selection bias [26,27,28,29].

Therefore, this study established a statistical model comprising a two-equations system, unconditional and conditional equations. For the unconditional equation, it was assumed that a dichotomous outcome variable $y_{1i}$ has a value of 1 if an individual i for i=1, 2, …, N has experienced healthcare needs and 0 if otherwise:$$y_{1i}=(y_{1i}{}^{*}=\beta_{1}\cdot X_{1i}+\varepsilon_{1i}>0)$$

For the conditional equation, it was assumed that among individuals experiencing healthcare needs, the quadchotomous outcome variable $y_{2i}$ has a value of j for j=1, 2, 3, 4, such that j=1 if the individual’s healthcare needs were met (the “met needs” group) and j=2, j=3, and j=4 if the individual’s healthcare needs were not met due to financial barriers (the “financial difficulty” group), time constraints (the “time constraint” group), and the lack of caring and support (the “lack of caring and support” group), respectively:$$y_{2ij}{}^{*}=\beta_{2j}\cdot X_{2i}+\varepsilon_{2ij}\;y_{2ij}=j\;\mathrm{if}\;y_{2ij}{}^{*}=\max\left(y_{2i1}{}^{*},\;y_{2i2}{}^{*},\;y_{2i3}{}^{*},\;y_{2i4}{}^{*}\right)$$

Here, the $X_{ki}\;\mathrm{for}\;k=1,\;2$ are the vectors of the explanatory variables, while the $\beta_{1}$ and $\beta_{2j}$ are the vectors of unknown coefficients. The error terms, $\varepsilon_{1i}$ and $\varepsilon_{2ij}$, were assumed to follow a multivariate normal distribution and were correlated with each other. This study chose people meeting their healthcare needs as the reference alternative for the quadchotomous outcome variable for the empirical analysis.

Regarding sample selection, rather than the Heckman two-step estimation model, this study employed a multivariable panel multinomial probit model with sample selection because several studies warned that Heckman’s estimator is likely to be influenced by multicollinearity between the estimated Mill’s ratio and explanatory variables [30,31,32]. In addition, the reason for using the multinomial probit model rather than the multinomial logit model in the second equation is that the former model can avoid the assumption of “the independence of irrelevant alternative property” in the latter model [33,34,35].

After establishing the statistical model, this study addressed seven statistical issues. First, Allison [36] noted that multicollinearity must be reduced even in multivariable analyses of a polychotomous outcome variable. Therefore, this study focused on the age variable around its mean, continually recategorizing each categorical explanatory variable and defining their reference categories differently. Consequently, the model did not exhibit considerable multicollinearity, as the variance inflation factor (VIF) values were < 2.61 in the univariate probit model in the first equation and < 2.35 in the multinomial probit model in the conditional second equation.

Second, the Hosmer–Lemeshow test was conducted to determine whether the first equation’s multivariable univariate probit model was well-fitted when the model had no survey weights. Third, because observations within the same individual are likely to be correlated, this study allowed for intra-individual correlation, relaxing the standard requirement that the observations must be independent, and then estimated the robust variance–covariance matrix corresponding to the parameter estimates. Fourth, this study attempted to find the best-fitted model by comparing the Akaike information criterion and Bayesian information criterion results with different sets of explanatory variables [37]. Fifth, this study examined all explanatory variables in the different models to find a model that satisfies the identification restriction [38,39,40,41]. As a result, for the identification, this study excluded some variables: the age-squared interaction-effect between the age and gender, and alcohol consumption in the first equation, as well as the active routine of physical exercise activity and obese variables in the second equation.

Lastly, only with the coefficient estimates and their corresponding standard errors for each explanatory variable, it may be difficult to understand an individual’s risk of experiencing either healthcare needs or a specific type of barrier to healthcare needs. Therefore, using the average marginal effects analysis method [42], this study estimated an average adjusted probability (AAP) (and their 95% confidence intervals [CIs]): (1) the probability that an individual would experience healthcare needs, (2) the probability that an individual would experience unmet healthcare needs, and (3) the probability that an individual would experience a specific type of barrier leading to unmet healthcare needs. Then, to facilitate ease of understanding, this study exhibited them in a graph by gender.

This study considered all characteristics to be time-varying (i.e., could change with time) and used the longitudinal weights provided by the KHP survey data for all longitudinal analyses. Statistical significance was set at *p* < 0.05. Statistical analyses were performed using SAS software (version 9.4; SAS Institute, Cary, NC, USA) and STATA 17 software (StataCorp, College Station, TX, USA).

## 4. Results

### 4.1. Percentage of People Experiencing Healthcare Needs and Each Type of Barrier to Healthcare

In this study’s sample, 96.7% of the participants reported that they had experienced healthcare needs in the last 12 months (Table 1). In addition, on average, 12.1% reported that they had experienced unmet healthcare needs due to healthcare barriers. The barrier that prevented people from meeting their healthcare needs most severely was “lack of caring and support” (5.3%), followed by “time constraint” (3.7%) and “financial difficulty” (3.1%), respectively.

### 4.2. Average Adjusted Probability of Experiencing Healthcare Needs across Age Groups by Gender and the Characteristics Associated with Healthcare Needs

As shown in Figure 1, this study obtained the AAP of experiencing healthcare needs across different age groups between 20 and 90 years by gender using the results from the multivariable panel multinomial probit selection model analysis (the first column in Table 2).

Regarding gender and age, the AAP of experiencing healthcare needs differed across age groups and genders. In men, this probability was U-shaped: it decreased from 93.8% at 20 years of age to 92.9% at 35 years of age and then increased to 99.6% at 90 years of age. In contrast, this probability continued to increase with age in women, reaching 95.7% at 20 years of age and 99.7% at 90 years of age. This probability was higher in women than in men in each age group.

Aside from gender and age, unmarried people were less likely than their married counterparts to experience healthcare needs (coefficient, −0.294; *p* < 0.001) (the first column in Table 2). People in the Seoul metropolitan area faced a lower risk of experiencing healthcare needs than those outside that area (coefficient, −0.254; *p* < 0.001). Compared to those who did not, people who completed college or had a higher level of education tended to be more likely to experience healthcare needs (coefficient, 0.064; *p* = 0.040).

MCA beneficiaries were more likely to experience healthcare needs than NHI beneficiaries (coefficient, 0.299; *p* = 0.005). Enrolling in private health insurance (coefficient, 0.083; *p* = 0.017) and functional limitations (coefficient, 0.437; *p* < 0.001) increased the risk of experiencing healthcare needs. Poor self-assessed health (coefficient, 0.332; *p* < 0.001) and having a chronic disease (coefficient, 0.418; *p* < 0.001) were highly associated with experiencing healthcare needs. In contrast, compared to current non-smokers, current smokers showed a lower risk of experiencing healthcare needs (coefficient, −0.122; *p* < 0.001).

Certain characteristics, such as occupation, household income, alcohol consumption, active physical exercise routine, and obesity status were not significantly associated with healthcare needs.

### 4.3. Average Adjusted Probability of Experiencing Each Type of Barrier That Leads to Unmet Healthcare Needs across Age Groups by Gender, and the Characteristics Associated with each Type of Barrier

This study obtained the AAP that an individual would experience unmet healthcare needs across different age groups between 20 and 90 years by gender, together with the probability that an individual would experience each type of barrier that leads to unmet healthcare needs (Figure 2) using the results obtained from the multivariable panel multinomial probit model with sample selection (the second, third, and fourth columns in Table 2).

For gender and age, in men, the AAP of experiencing unmet healthcare needs increased with age between 20 and 45 years and then decreased up to 90 years of age. In addition, the AAP of experiencing a specific type of barrier that leads to unmet healthcare needs differed across the age groups: the probability for the “financial difficulty” increased with age between 20 and 65 years and then decreased up to 90 years; the probability for the “time constraint” increased with age between 20 and 35 years and then decreased up to 90 years and the probability for the “lack of caring and support” increased with age between 20 and 75 years and then decreased up to 90 years.

Importantly, in men, the barrier that led most severely to unmet healthcare needs was not “financial difficulty” but either “time constraint” or “lack of caring and support” during adulthood—“time constraint” for those between 20 and about 50 years of age, and “lack of caring and support” for those between approximately 50 and 90 years of age.

In women, the AAP of experiencing unmet healthcare needs increased with age between 20 and 40 and then decreased up to 90 years of age. Further, concerning the AAP of experiencing a specific type of barrier that led to unmet healthcare needs, the probability of “financial difficulty” increased with age between 20 and 50 years and then decreased up to 90 years; the probability of “time constraint” increased with age between 20 and 30 years and then decreased up to 90 years; and the probability of “lack of caring and support” kept on increasing rapidly with age between 20 and 90 years.

Similar to men, in women, the barrier that led most severely to unmet healthcare needs was not “financial difficulty”. Rather, it was “time constraint” for those between 20 and approximately 40 years of age and “lack of caring and support” between approximately 40 and 90 years of age.

The adjusted associations between people’s experience of a specific type of barrier to healthcare needs and their characteristics, except for gender and age, are shown in Table 2 (the second, third, and fourth columns). This study emphasizes two points for an easy understanding of the results. First, the results are only for people who experienced healthcare needs because of the characteristics of the selection model. Second, the association between a specific characteristic of people and the people’s experience of each type of barrier is the result compared to the association between that characteristic and the people’s experience of meeting healthcare needs, because the latter was chosen as the reference alternative of the quadchotomous outcome variable.

Compared to married people, unmarried people were more likely to experience the “financial difficulty” (coefficient, 0.279; *p* < 0.001) and “lack of caring and support” (coefficient, 0.087; *p* = 0.049) barriers. Residents of Seoul metropolitan area, relative to non-residents, tended to frequently experience the “financial difficulty” (coefficient, 0.387; *p* < 0.001), “lack of caring and support” (coefficient, 0.173; *p* < 0.001), and “time constraint” (coefficient, 0.050; *p* = 0.019) barriers.

As for education level, compared to people who did not complete college or higher education, those who completed it had a lower risk of experiencing “financial difficulty” (coefficient, −0.225; *p* < 0.001). Regarding occupation, people with a blue-collar job were more likely than those with no job to experience the “time constraint” (coefficient, 0.477; *p* < 0.001) barrier. Compared to people with no job, those with a white-collar job were more likely to experience the “time constraint” (coefficient, 0.478; *p* < 0.001) barrier but less likely to experience the “lack of caring and support” (coefficient, −0.251; *p* < 0.001) barrier. In terms of household income, compared to people belonging to the lowest quintile group, people belonging to the medium group were much less likely to experience the “financial difficulty” (coefficient, −0.579; *p* < 0.001) barrier and less likely to experience the “lack of caring and support” (coefficient, −0.138; *p* = 0.002) barrier. In addition, people belonging to the highest quintile group had a lower risk of experiencing any of these three types of barriers: “financial difficulty” (coefficient, −1.178; *p* < 0.001), “lack of caring and support” (coefficient, −0.164; *p* = 0.004), and “time constraint” (coefficient, −0.077; *p* = 0.043).

As for the status of national health security programs, compared to NHI beneficiaries, MCA beneficiaries exhibited a higher risk of experiencing a “lack of caring and support” (coefficient, 0.203; *p* = 0.015). Concerning a private health insurance plan, people holding it had a reduced risk of experiencing “financial difficulty” than those who did not (coefficient, −0.152; *p* = 0.001).

Compared to people without a functional limitation, those with it appeared to have a much higher risk of experiencing a “lack of caring and support” (coefficient, 0.655; *p* < 0.001) and a higher risk of experiencing “financial difficulty” (coefficient, 0.389; *p* < 0.001). Compared to current non-smokers, current smokers were more likely to experience “financial difficulty” (coefficient, 0.234; *p* < 0.001), “lack of caring and support” (coefficient, 0.164; *p* = 0.002), and “time constraint” (coefficient, 0.090; *p* = 0.011). Compared to non-consumers, consumers of alcohol had a higher risk of experiencing “time constraint” (coefficient, 0.116; *p* = 0.002) and of experiencing “lack of caring and support” (coefficient, 0.078; *p* = 0.028). Compared to people not reporting that their health is poor, people reporting that their health is poor were more likely to experience “financial difficulty” (coefficient, 0.634; *p* < 0.001), “lack of caring and support” (coefficient, 0.453; *p* < 0.001), and “time constraint” (coefficient, 0.312; *p* < 0.001). Compared to people without a chronic disease, those with a chronic disease exhibited a lower risk of experiencing “lack of caring and support” (coefficient, −0.156; *p* < 0.001).

## 5. Discussion

### 5.1. Characteristics Associated with Experiencing Healthcare Needs

Identifying the characteristics associated with experiencing healthcare needs can assist in identifying people who are likely to be in need of healthcare. For ages between 35 and 90 years under consideration, this study shows that people who were likely to experience healthcare needs were women relative to men—in men, those who were younger than 35 years of age and those who were older than this age; whereas in women, those who were older (Figure 1).

In addition to gender and age, the characteristics of individuals associated with experiencing healthcare needs were being married, non-residents of the Seoul metropolitan area, having a college education or higher, being an MCA beneficiary, being enrolled in a private health insurance plan, having a functional limitation, being a current non-smoker, having poor self-assessed health, and having a chronic disease (Table 2).

Compared to younger people, older people seem to need more healthcare services because they, particularly older women, care more about their health conditions and visit physicians more often [13]. Interestingly, however, in men between the ages of 20 and 35, age showed a negative relationship with healthcare needs. This may be because men in this age group in Korea are too preoccupied with jobs to feel their need for healthcare services [43,44].

Marriage may increase healthcare needs through the spouse’s care and his (or her) encouragement to visit a physician [45,46]. One reason why people residing in the Seoul metropolitan area are less likely to experience healthcare needs than those living outside this area may be that these residents have continued to meet sufficient healthcare needs because healthcare providers are highly concentrated in this area [47,48]. Moreover, people may be able to manage their health conditions better with more information about health in the Seoul metropolitan area, thereby not leading to healthcare needs.

More education seems to alert people to control their health behaviors or to receive timely healthcare services [49,50]. It may be postulated that a private health insurance plan increases healthcare needs because people who buy it are more susceptible to health problems than those who do not [51]. Compared to current smokers, people who are not current smokers (never or ex-smokers) may be more responsive to health problems [52]. People who have functional limitations [53], poor self-assessed health [54], and chronic disease [55] have a higher likelihood of having healthcare needs than their respective counterparts.

### 5.2. Characteristics Associated with Experiencing a Specific Type of Barrier to Healthcare

This study found that different people may experience different barriers preventing them from meeting their healthcare needs (Table 2 and Figure 2). Meanwhile, identifying the characteristics of people associated with experiencing a specific type of barrier can help researchers search for people who are very likely to suffer from this barrier and help policymakers develop policies to reduce it.

The adjusted predicted probability of experiencing a specific type of barrier that leads to unmet healthcare needs differed across gender and age groups (Figure 2). In men, the three probabilities of experiencing “financial difficulty”, “time constraint”, and “lack of caring and support” increased with age and after a certain age decreased with age; in women, the two probabilities of experiencing “financial difficulty” and “time constraint” revealed a similar pattern. However, women’s probability of experiencing a lack of caring and support increases rapidly with age. These findings differ from those of previous studies emphasizing a monotonic pattern with age: system barriers, such as financial barriers, are positively associated with age [56], whereas personal factors, such as time constraints, are negatively associated with age [56,57,58,59].

Concerning “financial difficulty”, compared to the association between people’s characteristics and the people’s experience of meeting healthcare needs, people’s characteristics that were positively associated with those experiencing this barrier were as follows: being unmarried, living in the Seoul metropolitan area, not having a college or higher education, being in neither the medium nor the highest quintile of household income, not having a private health insurance plan, having a functional limitation, being a current smoker, and reporting poor self-assessed health.

These characteristics seem related to people’s low socioeconomic status and health-or function-related status [14,15,58,60,61]. Particularly, the finding that residents in the Seoul metropolitan area are more likely to experience “financial difficulty” than non-residents may be partly because healthcare providers are highly concentrated in the Seoul metropolitan area and under a high degree of competition. To reach their target income, healthcare providers may induce people to visit them more frequently, probably by offering high-quality and high-price healthcare services not covered by the NHI program [9,47,48].

As for the “time constraint”, relative to the association between people’s characteristics and the people’s experience of meeting healthcare needs, the people’s characteristics that were positively associated with their experiencing the “time constraint” were as follows: residing in the Seoul metropolitan area, having a job in a labor market (a blue- or white-collar job), not being in the highest quintile of household income, being a current smoker, being an alcohol consumer, and reporting poor self-assessed health. These characteristics seem to be related to the lack of time to visit a healthcare facility either because of living conditions or workplace environments [62,63,64]. This study’s findings align with those of prior studies in concluding that employed people use healthcare services less frequently than unemployed people [65,66,67].

A previous study in Korea revealed that, compared to economically inactive people, economically active people received healthcare services less often due to “time constraint” [68]. Considering that the average annual hours worked per worker in Korea in 2021 is 1915 h, which is longer than the average value of the OECD countries, 1716 h [69], time constraints in workplaces seem to make it difficult for people to visit a healthcare facility when necessary. In addition, economically active people consume tobacco and alcohol to forget their health problems even when they feel that they need healthcare services because of high stress in a competitive society like Korea and time constraints, whereby they easily forfeit their visit to a healthcare facility [70,71].

Regarding “lack of caring and support”, compared to the association between people’s characteristics and their experience of meeting healthcare needs, people’s characteristics that were positively associated with experiencing it were as follows: being unmarried; living in the Seoul metropolitan area; not having a white-collar job; being in neither the medium nor the lowest quintile of household income; being a Medical Care Aid beneficiary; having a functional limitation; being a current smoker; being an alcohol consumer; reporting poor self-assessed health; not having a chronic disease.

Unfortunately, there are no studies with findings comparable to those of this study because no study has ever explored the characteristics associated with the “lack of caring and support”, which are conditional on the existence of healthcare needs. Although the barriers are neither conditional on the existence of healthcare needs nor are they divided into three types, one recent study in Korea revealed that non-financial barriers remained more likely than financial barriers to be associated with unmet healthcare needs during adulthood in both men and women [9]. This previous study’s result is consistent with the present study’s finding that non-financial barriers are more likely than financial barriers to prevent people from meeting their healthcare needs. More specifically, this study showed that in both men and women, the barrier most likely to cause people’s healthcare needs not to be met was “time constraint” in an early stage of the life cycle of adulthood and “lack of caring and support” in its late stage.

### 5.3. Policy Issues, Policy Goals, and Policy Suggestions

Based on the results obtained earlier, this section presents four policy issues that Korea must consider to reduce unmet healthcare needs. Further, it sets three policy goals to overcome these issues and discusses the direction of reorganizing the management and operation of the national health security system to achieve these goals.

#### 5.3.1. Policy Issue 1: No Healthcare Professional Assists in Whether People Should Visit a Healthcare Facility

Most healthcare needs are subjective and not objective. Whenever people feel sick, they usually try to obtain an objective opinion from a healthcare professional about whether to visit a healthcare facility and undergo medical examinations or treatments.

The system that most countries are equipped with for such people is primary care, which has been regarded as an essential element of universal health coverage [17,72,73,74,75,76]. Therefore, countries that lack primary care are trying to expand their coverage to include the entire population [77]. In these countries, when people are sick, they first come into contact with the healthcare system through primary care, which provides, as one of its functions, consistent long-term counseling and management to those enrolled in primary care physicians [3].

Korea is considered an exceptional country because its universal health coverage is not based on primary care. Because of the critical lack of primary care physicians, it is almost impossible for people to receive a healthcare professional’s objective opinion regarding whether they need to visit a healthcare facility. Whenever people are sick, they usually experience a surge of panic and set out on a long journey to directly visit a variety of specialist clinics, secondary care hospitals, or tertiary care hospitals, because all of them are allowed to provide outpatient care as people’s first contact with a physician.

People tend to visit healthcare facilities as often as possible until the fear of illness or death disappears; therefore, the frequency of visits to healthcare facilities increases more through the moral hazard phenomenon, as public or private health insurance covers more healthcare expenses. In addition, if the desire to visit healthcare facilities several times is not realized enough to feel that they are in good health, their healthcare needs are likely not met. As a result, Korea is known to have the highest number of visits to a doctor among OECD member countries [3]. Nonetheless, the proportion of people experiencing unmet healthcare needs is higher in Korea than that in most European countries [9].

Therefore, this study intends to recommend that, as in the healthcare system of other advanced countries, Korea’s healthcare system should be equipped with primary care physicians who will provide people who are sick with objective opinions on whether to visit healthcare facilities or self-care at home.

#### 5.3.2. Policy Issue 2: Even When People Should Visit a Healthcare Facility, None of the Healthcare Professionals Assists People in Visiting It

After realizing the need to visit healthcare facilities, people will want to know (1) if, despite feeling that symptoms are getting better or are not serious now, they should visit a healthcare facility, (2) what type of healthcare facilities to visit, (3) what to do when the reservation wait is too long, and (4) if people should visit healthcare facilities without their regular doctors.

In Korea, the universal healthcare system does not link people with healthcare professionals to provide such information. Therefore, sick people tend to inevitably go from place to place to cure their illnesses. Specialist clinics, secondary care hospitals, and tertiary care hospitals offer both outpatient and inpatient services to patients, so they compete fiercely to get more people to visit and stay with them more often. The possible reasons for strengthening this are (1) they are mostly privately owned; (2) most healthcare services are reimbursed according to the fee-for-service payment schedule, whether they are outpatient or inpatient care services, and whether they are covered by public or private health insurance; (3) there is no defined catchment area for healthcare utilization; (4) the referral system is in name only, not well established; and (5) healthcare information systems for patients are not interconnected among healthcare facilities. Meanwhile, if the waiting time for a physician for whom people search for themselves is long, they usually keep waiting for the physician without finding another option, or they give up their examination or counseling, which puts them at risk of poorer health.

Therefore, it is necessary to establish a framework for receiving objective opinions from healthcare professionals on what to do when the situation is severe enough that people visit health facilities. Consequently, this study recommends that Korea’s healthcare system, like that of most advanced countries, secure a sufficient number of primary care physicians and provide the public with the necessary healthcare-related information.

#### 5.3.3. Policy Issue 3: Even When People Should Visit a Healthcare Facility, People Cannot Secure the Time to Visit It

To receive the necessary healthcare services, people ought to spend their time commuting to a healthcare facility, wait there, and receive an examination or treatment. For the working population, it is often difficult to secure time to receive timely and appropriate healthcare services in order not to worsen their health. This is especially the case for self-employed people, whose proportion is very high in Korea compared to other OECD countries [78]. When visiting a healthcare facility, they tend to have difficulties committing their work to others.

This study recommends that social policies be established to guarantee relatively flexible schedules to help the working population receive timely and appropriate healthcare services. In addition, onsite workplace clinics need to be set up, possibly supported by the earmarked subsidy of the government, for workers at medium- or large-scale workplaces [79,80]. Afterhours healthcare services are recommended to be expanded and strengthened for workers at small-scale workplaces and the self-employed [81,82,83].

Meanwhile, the recent development of digital health services may be useful for people who lack time to visit a healthcare facility, thereby ascertaining better access to healthcare [3]. For example, consultations with primary care physicians can occur through teleconsultation methods, such as traditional telephone, online, and internet phone-like video calls [3,84]. Korea’s highly developed IT industry will contribute greatly to the healthcare system. Thus, with the rapid increase in access to computers and mobile phones, telemedicine and web-based health services will have to be encouraged to participate in many areas of the healthcare system in keeping with the development of information protection methods [73,85,86].

In addition, even now, the nation’s healthcare system must encourage direct or internet visits by healthcare professionals. To do so, Korea must escape the dilemma of possessing a serious lack of primary care physicians for a long time. Otherwise, it may face great difficulties in providing healthcare services to patients with chronic diseases and functional limitations, which will increase with the rapid aging of the population.

#### 5.3.4. Policy Issue 4: Even When People Who Have to Visit a Healthcare Facility Have Difficulty Visiting It, None of the Social Service Professionals Help People Visit It

To receive necessary healthcare services, people usually face various healthcare barriers that concern social services. They are (1) financial difficulty, (2) functional limitation or poor health, (3) transportation challenges to reach a healthcare facility, and (4) difficulty in visiting a healthcare facility because they have to take care of children. Unfortunately, Korea’s social service provision system is not well-connected to the healthcare system.

Regarding financial difficulties in healthcare spending, that is, “financial difficulty” in this study, people are burdened with their own medical expenses even though all people are covered by the national health security system and most of them have at least one private health insurance plan.

One of the main reasons may be that a considerable number of non-essential healthcare services not covered by the national health security system are provided to the public simultaneously, along with essential healthcare services that are covered by this system. The government does not regulate the provision and prices of these non-essential healthcare services. Therefore, as the national health security system covers more healthcare services and their prices are more tightly regulated, healthcare facilities (mostly privately owned) are likely to have incentives to raise the price of non-essential healthcare services and offer these services more to patients, which is like a balloon effect.

According to a previous study, the number of physician consultations differs across socioeconomic groups. For example, compared to people in the highest income quintile, those in the lowest income quintile are less likely to visit a physician to meet their healthcare needs [3]. Therefore, if all people are registered with primary care physicians, who acknowledge their patients’ financial difficulties with medical expenses, the physicians may be able to discourage them from receiving expensive but non-essential healthcare services. Alternatively, they may introduce their patients to social service professionals who can help such patients reduce their financial burden through public or private assistance programs [87].

Regarding non-financial barriers to healthcare, prior studies have reported that functional limitations in people exacerbate their ability to access healthcare [88,89,90,91,92]. According to a recent study in Korea, people with functional limitations have a substantial high risk of experiencing non-financial barriers to healthcare [9]. Faced with Korea’s rapid population aging and the subsequent increase in the population with functional limitations [93], the problem of the increasing unmet healthcare needs of people with functional limitations will be of great concern in the nation’s healthcare system. Therefore, Korea should greatly reform its national health security system to improve healthcare access for people with functional limitations. Thus, it is necessary to prepare a framework for primary care and link it to the provision of social services. This will facilitate visits of people with functional limitations to health facilities [53,94,95] and reduce financial barriers to their access to healthcare by maintaining continuity, inclusiveness, and coordination for people with functional limitations [96].

It would be possible for individuals to receive timely and adequate healthcare services when needed if primary care providers and social service providers in each community combine to form various types of “primary care provider networks” to cater to the healthcare needs of individuals. For example, social services provided by the “primary care provider networks” will enable many people to visit healthcare facilities, including those with financial difficulties, functional limitations, transportation problems to travel to healthcare facilities, and those who have difficulty visiting them because of child caring [97,98,99,100,101].

In addition, strengthening access to primary care and social service providers in each community can positively affect patients and the healthcare system. For example, in countries such as the Netherlands, France, and Norway, hospital stays can be reduced by increasing the capacity of intermediate care facilities and home care [3,17,102]. Meanwhile, for mental health treatment and support, the OECD recommends establishing a whole-of-society approach in which healthcare should be strongly integrated with social welfare, labor, and youth policies [3,103].

#### 5.3.5. Policy Goals for Implementing the Recommendations on the Four Policy Issues

This study has proposed recommendations for solving the four policy issues mentioned above. The recommendations seem to be implemented effectively under three policy goals: (1) strengthening primary care, (2) establishing community-based integrated care, and (3) building up people-centered care.

First, concerning primary care, its key aim is to induce primary care physicians to assist people to maintain their health well by (1) providing health promotion services, (2) preventing diseases, (3) providing a consistent long-term point-of-care; (4) treating common (or uncomplicated) conditions, (5) managing chronic conditions, (6) tailoring and coordinating care for those with multiple healthcare needs, (7) referring patients to hospital-based services when appropriate, (8) supporting patients’ self-management of their conditions, and (9) managing new health complaints [3]. According to prior studies, good primary care is known to exert a positive influence on efficiency, equity, and responsiveness in the nation’s healthcare system; hence, improving people’s health by making better use of healthcare resources, reducing socioeconomic inequalities in health, and making the health system people-centered [3,17].

As previously mentioned, Korea has not long focused on primary care. Consequently, it possesses a significant shortage of primary care physicians. Therefore, people with undiagnosed health problems and those in need of continuous management of chronic diseases should visit specialists working in clinics, secondary care hospitals, or tertiary care hospitals [93].

Surprisingly people in Korea often regard the first-contact treatment provided by specialists as “primary care”, perhaps because of misinformation. However, according to the OECD, primary care is defined as a range of services (commonly referred to as “basic care services”) covering (1) general outpatient care, (2) general dental care, (3) home-based curative care conducted through home visits by general practitioners (GPs) or nurses, and (4) prevention services provided by ambulatory care providers. Such services are excluded from primary care if they are provided in a hospital or by an outpatient specialist [3].

Although it seems late, Korea needs to equip and empower its health security system with an adequate number of primary care physicians through various policy strategies, including an increased supply of primary care physicians as well as an appropriate level of training and remuneration for them.

It is worth considering the cases of other advanced countries if it is expected to be difficult or take much time. Compared to the number of doctor consultations per capita in Korea, those in Canada, Finland, Ireland, New Zealand, Sweden, and the UK are relatively lower. This is partly because nurses and other health professionals play an important role in primary care, for example, for people with chronic diseases and those with minor health issues, thereby reducing the need for doctor consultations [3,104]. Therefore, extensive education in primary care, comparable to that of nurse practitioners and physician assistants in the USA, may expand the practice scope of Korea’s registered nurses in primary care areas as that of their counterparts in the USA [105,106,107,108,109]. To some degree, the rapid development of the digital health sector will contribute to solving the lack of primary care physicians if digital health improves the productivity of primary care physicians [110,111,112].

If primary care is strengthened in Korea, primary care providers should communicate with their enrollees about their healthcare needs and facilitate the coordination of a variety of care for their enrollees in a community. Consequently, limiting unnecessary visits to health facilities and hospitalizations will improve residents’ health outcomes and reduce wasteful spending. As such, primary care may exert a constructive effect on establishing community-based integrated care and building up people-centered care.

As for community-based integrated care, the importance of national efforts in Korea cannot be overemphasized. One of the reasons for this is that Korea has gained notoriety for its fragmentary healthcare delivery system. In addition, as Korea is facing rapid population aging, it will be inevitable to promote the long-term transition of health care and social care, as well as improve the interaction among providers in the community and efficiently use all available resources. Previous studies [3,17,113,114] have shown that community-based integrated care can improve health performance and promote the sharing of health-enhancing methods. It also increases monetary value by improving coordination while reducing redundant and unnecessary care. Without proper care integration between health, long-term, and social care, the process of helping people with complex health conditions—especially mental illnesses—receive adequate healthcare services cannot be performed well.

Many advanced countries are gradually making efforts to reflect people’s opinions in their healthcare systems actively. For example, by developing and monitoring patient-reported measures, they strive to improve healthcare quality and develop a people-centered healthcare system. In many countries, healthcare facilities are responsible for regularly measuring and reporting patient experience data obtained when they receive healthcare, and governments are trying to establish and provide standardized procedures for analyzing such data and reporting its results [3]. In Korea, the healthcare utilization rate, the proportion of adults reporting unmet healthcare needs, the proportion of adults rating their health as bad or very bad, and deaths by suicide per population are the highest among the OECD member countries. However, the number of physicians per population, particularly primary care physicians, is the lowest [3]. In this bizarre universal health security system, it seems urgent to establish the direction of people-centeredness and reform its system one step at a time in that direction. In addition, because the population is aging most rapidly among OECD member countries, healthcare systems in Korea will need to adapt to meet the needs of an aging population, which is likely to include greater demand for long-term care and a greater need for integrated, people-centered care [3,113].

#### 5.3.6. Reform Directions of the National Health Security System Governance Structure to Achieve the Three Policy Goals

The governance structure of Korea’s health security system needs to be widely restructured to achieve the three policy goals mentioned above.

The NHI in Korea is managed by a huge single public insurer, the NHIS. All funds are collected in it, and everyone, regardless of sociodemographic differences, is enlisted in it, indicating a high level of income redistribution [115]. However, unlike most developed countries’ public healthcare systems, decentralized management and operations are almost nonexistent, as the central authority controls most of them without allowing the autonomy of regional (or local) authorities.

This inflexible system can reduce interregional inequality in financial barriers to healthcare services covered by the NHI. However, it can also create problems that strengthen interregional inequality in non-financial barriers. This is because non-financial barriers depend strongly on the community population and health-related characteristics. As a result, access to healthcare may vary significantly from region to region due to interregional differences in financial barriers to healthcare services not covered by the NHI and interregional differences in non-financial barriers. Recent studies have shown that inter-regional disparities in health are significant [116,117].

Therefore, considering the results of both a recent study [9] and this study, which reveal that non-financial barriers mostly cause unmet healthcare needs in Korea, Korea’s healthcare system needs to be reformed to play different roles between central and regional authorities as well as ensure that each regional authority is responsible for its residents’ health.

For example, the NHI’s central authority must establish a common part of the framework with which all regional authorities will comply. This should include the enhancement of (1) primary care, (2) community-based integrated care, and (3) people-centered care, whereby each regional authority should help ensure that their residents receive adequate healthcare services on time. The key contents that the central authority needs to include in the common body of the framework for regional authorities are (1) reforming the governance of a regional authority, (2) establishing financing sources, (3) excavating care needs, (4) building up a skilled workforce and necessary care providers, (5) organizing care delivery, (6) enhancing access to care, (7) developing strong information systems, and (8) designing payment mechanisms to ensure aligned financial incentives [3,118].

Meanwhile, the central authority may develop and propose various governance models, allowing each regional authority to choose the best fit for their residents. Furthermore, the central authority may listen to the difficulties of regional authorities and solve them, monitor their activities, evaluate their outcomes, provide awards and penalties to the management of regional authorities, and make them public to facilitate “internal competition” among regional authorities. One of the reasons is that the most important thing for the central authority is to encourage local authorities to make great efforts to transparently, equitably, and efficiently improve the health of their residents. Regarding financial barriers to healthcare, the central authority that collects all financial resources may allocate resources to regional authorities according to a risk-adjusted, capitated fee schedule so that all regional residents can receive the “common benefits” set by the central authority without much financial difficulty.

Next, each regional authority should establish a region-specific part of the framework to be added to the common part established by the central authority. This is because the health-related characteristics of residents, providers, and environments differ from region to region. In addition, regional authorities could focus effectively on reducing non-financial and financial barriers to healthcare because, compared to the central authority, they can be more knowledgeable about community resources in their region and mobilize them better to improve people’s access to healthcare.

Compared to the central authority, regional authorities can work more effectively in linking primary care professionals with social or long-term care professionals; hence, saving money provided by the central authority and providing region-specific benefits to the population. In addition, they can endeavor harder than anyone else to establish educational and training facilities for primary, social, and long-term care professionals in their regions to supply scarce manpower. Consequently, by adopting and utilizing their appropriate mechanisms, each regional authority will be able to provide fairly the necessary healthcare services to each regional resident and reduce different barriers to healthcare.

The best advantage of this “managed approach toward decentralization” is that all regional authorities will compete with each other for transparent, fair, and efficient delivery of healthcare. Moreover, those with low performance will emulate those with high performance, so Korea can simultaneously conduct “an incremental reform” to improve its healthcare system.

Regarding strengthening people-centered care, the role of the central authority seems more crucial than that of regional authorities. For example, the central authority could make the findings about people-centered care public both from subjective and objective points of view, so that it can periodically publish the so-called “regional health report” as Korea’s version of “OECD Health at a Glance.”

From a subjective perspective, the central authority can enforce periodic health surveys for residents and healthcare providers in each region to detect and report the full range of healthcare system problems, including the prevalence rates of all types of barriers to healthcare and people’s satisfaction with each healthcare service. From an objective point of view, the central authority can analyze vast databases covering the entire population and report its results by region. Such databases are (1) qualification database for NHI and MCA beneficiaries; (2) treatment database for treatment details; (3) type of disease for details of prescription on the data from medical care, dental care, oriental medicine institution, and pharmacy; and (4) medical checkup database.

In addition, concerning people-centered care, the central authority must remember that providing people access to their data and information on their health conditions and the contents of examinations and treatments is a key factor in a people-centered healthcare system. According to a previous study, people with low sociodemographic status, such as the elderly, people with lower education, and those from poor households, had more difficulty searching for health-related information online [84]. Therefore, to develop a people-centered healthcare system, the central authority must work together with other government departments to make great efforts to reduce the health and digital health literacy of people with low sociodemographic status [3].

### 5.4. Limitations and Future Research

First, the KHP survey data on unmet healthcare needs and reasons analyzed in this study were self-reported information, which may have included recall bias. However, because individuals can provide more accurate information about themselves, previous studies in many countries have used self-reported data related to unmet healthcare needs [6,56,58,59,119,120,121,122,123,124]. Second, in the KHP survey, respondents were asked to identify only one of the most important reasons why their healthcare needs had not been met in the past 12 months. Therefore, this survey could not identify all barriers that prevented respondents from accessing healthcare.

Third, owing to a lack of information, this study could not include characteristics, such as social capital and social support [13,125], as well as the emotional satisfaction of consumers toward healthcare services [126], in the analysis. Fourth, because an increasing number of foreigners are marrying Koreans than in the past, it would be interesting to include characteristics, such as ethnicity, immigrant status, and religion in analyzing unmet healthcare needs and why [57,60]. However, the KHP survey data did not contain this information.

Fifth, if a doctor recommends a patient to receive the necessary healthcare services, but the patient does not receive them, the patient may face a risk of a deteriorating health condition compared to the case where there was no doctor’s recommendation. Unfortunately, no such information in the data was used in this study. However, if this information can be obtained later, a more rigorous study can be conducted considering whether the doctor recommends healthcare services.

Sixth, before conducting the research, this study considered incorporating the Andersen behavioral model [127]. However, it seemed that the Andersen behavioral model was not appropriate for this study, unlike other studies [128,129]. One reason was that whereas the Andersen behavioral model focused on actual healthcare utilization, this study aimed to analyze both the “subjective” unmet healthcare needs (“needs” and “non-needs”) and the “subjective” outcome of the “subjective” unmet healthcare needs (“met needs”, “financial difficulty”, “time constraint”, and “lack of caring and support”). Another reason was that the second outcome variable in this study, the “subjective” outcome of the “subjective” unmet healthcare needs, was a combination of (1) the “subjective” outcome of the “subjective” unmet healthcare needs and (2) independent variables (access-enabling factors like availability of health providers, transportation, and one’s ability to pay for healthcare).

Seventh, regarding the trajectories of Figure 1 and Figure 2, this study used “standard growth analysis” to focus more on the study purpose and present various policy implications based on the results of this study. Future studies exploring different trajectories across heterogeneity between individual groups in temporal variations of specific variables such as unmet healthcare needs will use “latent class growth trajectory modeling” [130,131].

Eighth, changes in unmet healthcare needs and barriers to healthcare across ages may vary by different sociodemographic characteristics. Future research will be performed to stratify the population by a specific characteristic to identify a sub-population targeted to efficiently reduce unmet healthcare needs. Ninth, because this study used a longitudinal dataset, attrition bias would arise if people’s unmet healthcare needs and their healthcare access outcomes were correlated with a sample exit. However, the attrition rate in this study was relatively small. The 2014 survey includes 15,379 respondents aged 19 or more. Over time, attrition reduced the sample by approximately 2 percentage points per wave so that, in 2018, 92.7% (14,261) of the initial respondents remained in the sample.

Finally, Korea’s national health insurance system is based on universal health insurance systems in European countries, so there will be much to learn from the development trajectory of their systems. For example, strategies to effectively minimize unmet healthcare needs in Korea can be derived by analyzing the experiences of European countries. Therefore, it seems necessary to regularly conduct health surveys of common content in cooperation with the European Union to compare the actual conditions of unmet healthcare needs and the policy performance of each country to reduce them [9]. Since there are no such data, this study could not conduct comparative studies with European countries.

## 6. Conclusions

Until now, healthcare systems worldwide have focused their attention on the problem of people not receiving the healthcare services they want because of insufficient money. Even countries with universal health coverage seem to keep paying policy attention to the financial barriers. However, rigorous research on barriers to access to healthcare may call for shifting more attention to non-financial barriers than financial barriers, which may be an obstacle to meeting healthcare needs.

This study used recent longitudinal data and advanced statistical analysis methods to show that, in Korea, the main causes of unmet healthcare needs faced by adults are not financial difficulties but time constraints up to a certain age and the lack of caring and support after that age. Based on these results, this study implies that if the government’s efforts to reduce unmet healthcare needs focus only on lowering financial barriers, it will only require excessive government finances and fail to meet its intended purpose. In the current situation in Korea, this study proposes a policy to secure the foundation of “primary care” that is scarce in Korea and expand it to “community-based integrated care” and “people-centered care.” To this end, this study recommends that the current excessively centralized national health insurance organization be decentralized in terms of management and operation to plan and implement regional-specific policies to lower non-financial barriers while maintaining financial integration.

The findings and recommendations will help countries with universal health coverage similar to Korea, where the population ages rapidly, the number of people with chronic diseases and functional disabilities increases, and where people experience unmet healthcare needs more often due to non-financial barriers than due to financial barriers.

## Figures and Tables

**Figure 1 healthcare-10-02243-f001:**
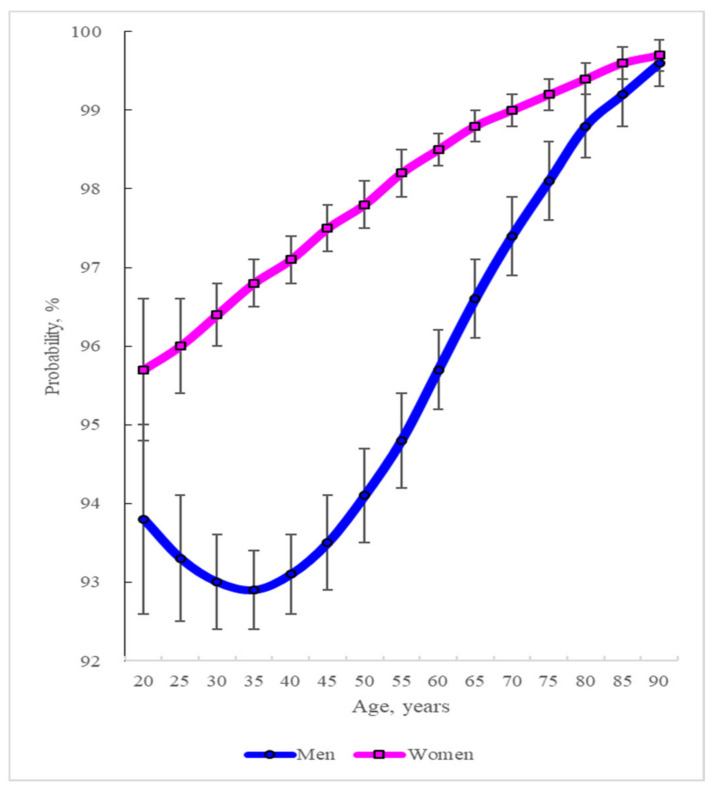
Change in the average adjusted probability (and its 95% confidence interval) of people’s experience of healthcare needs across different age groups by gender. Source: The Korea Health Panel survey data (2014–2018).

**Figure 2 healthcare-10-02243-f002:**
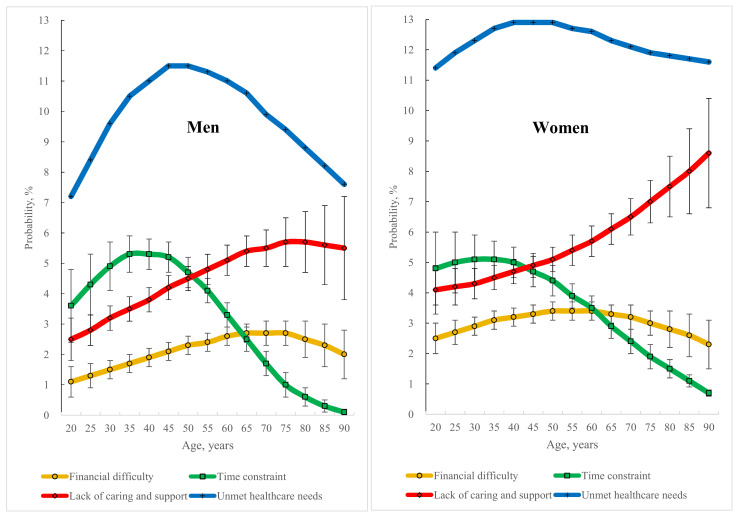
Change in the average adjusted probability (and its 95% confidence interval) of people’s experience of a type of barrier that leads to unmet healthcare needs across different age groups by gender. Source: The Korea Health Panel survey data (2014–2018).

**Table 1 healthcare-10-02243-t001:** Sample characteristics and their descriptive statistics for each year.

Characteristics	2014 (*n* = 14,691)		2015 (*n* = 14,001)		2016 (*n* = 13,471)		2017 (*n* = 13,408)		2018 (*n* = 13,359)		Total (*n* = 68,930)	
	Mean	SD	Mean	SD	Mean	SD	Mean	SD	Mean	SD	Mean	SD
Healthcare needs	0.970	(0.170)	0.970	(0.169)	0.970	(0.169)	0.948	(0.222)	0.976	(0.154)	0.967	(0.178)
Unmet healthcare needs	0.128	(0.334)	0.137	(0.344)	0.111	(0.314)	0.106	(0.308)	0.121	(0.326)	0.121	(0.326)
Financial difficulty	0.033	(0.180)	0.038	(0.192)	0.031	(0.173)	0.026	(0.160)	0.025	(0.155)	0.031	(0.173)
Time constraint	0.040	(0.196)	0.039	(0.193)	0.031	(0.173)	0.035	(0.183)	0.041	(0.197)	0.037	(0.189)
Lack of caring and support	0.054	(0.227)	0.060	(0.238)	0.049	(0.216)	0.045	(0.208)	0.056	(0.230)	0.053	(0.224)
Sociodemographics												
Man	0.466	(0.499)	0.464	(0.499)	0.460	(0.498)	0.460	(0.498)	0.460	(0.498)	0.462	(0.499)
Age (years)	51.616	(17.264)	52.274	(17.474)	52.971	(17.628)	53.350	(17.844)	53.687	(18.089)	52.753	(17.668)
Unmarried	0.309	(0.462)	0.316	(0.465)	0.324	(0.468)	0.332	(0.471)	0.341	(0.474)	0.324	(0.468)
Seoul metropolitan area	0.379	(0.485)	0.376	(0.484)	0.374	(0.484)	0.376	(0.484)	0.376	(0.484)	0.376	(0.484)
College or higher	0.306	(0.461)	0.309	(0.462)	0.313	(0.464)	0.322	(0.467)	0.331	(0.471)	0.316	(0.465)
Occupation												
No job	0.397	(0.489)	0.418	(0.493)	0.405	(0.491)	0.401	(0.490)	0.391	(0.488)	0.402	(0.490)
Blue-collar job	0.414	(0.493)	0.393	(0.488)	0.402	(0.490)	0.403	(0.491)	0.410	(0.492)	0.404	(0.491)
White-collar job	0.190	(0.392)	0.190	(0.392)	0.193	(0.395)	0.195	(0.397)	0.199	(0.399)	0.193	(0.395)
Household income												
Lowest quintile	0.201	(0.401)	0.200	(0.400)	0.200	(0.400)	0.200	(0.400)	0.200	(0.400)	0.200	(0.400)
Medium	0.599	(0.490)	0.600	(0.490)	0.600	(0.490)	0.603	(0.489)	0.600	(0.490)	0.601	(0.490)
Highest quintile	0.200	(0.400)	0.200	(0.400)	0.200	(0.400)	0.197	(0.398)	0.199	(0.399)	0.199	(0.399)
Medical Care Aid	0.032	(0.175)	0.033	(0.178)	0.035	(0.185)	0.033	(0.180)	0.033	(0.178)	0.033	(0.179)
Private health insurance	0.676	(0.468)	0.705	(0.456)	0.718	(0.450)	0.733	(0.442)	0.744	(0.436)	0.715	(0.452)
Physical and health conditions												
Functional limitation	0.059	(0.236)	0.079	(0.269)	0.069	(0.253)	0.061	(0.239)	0.064	(0.244)	0.066	(0.249)
Current smoker	0.207	(0.405)	0.182	(0.386)	0.180	(0.385)	0.175	(0.380)	0.170	(0.376)	0.183	(0.387)
Alcohol consumer	0.645	(0.479)	0.656	(0.475)	0.651	(0.477)	0.653	(0.476)	0.656	(0.475)	0.652	(0.476)
An active routine of physical exercise activity	0.383	(0.486)	0.394	(0.489)	0.392	(0.488)	0.375	(0.484)	0.353	(0.478)	0.380	(0.485)
Obese	0.245	(0.430)	0.249	(0.432)	0.254	(0.435)	0.272	(0.445)	0.272	(0.445)	0.258	(0.438)
Poor self-assessed health	0.840	(0.366)	0.858	(0.349)	0.851	(0.356)	0.849	(0.358)	0.848	(0.360)	0.849	(0.358)
Chronic disease	0.626	(0.484)	0.631	(0.483)	0.641	(0.480)	0.623	(0.485)	0.630	(0.483)	0.630	(0.483)

Note: SD denotes standard deviation. All characteristics were considered to be time-varying. Source: The Korea Health Panel survey data (2014–2018).

**Table 2 healthcare-10-02243-t002:** The associations between people’s characteristics and the people’s experience of healthcare needs, and between people’s characteristics and the people’s experience of a specific type of barrier that leads to unmet healthcare needs among the people who needed healthcare services.

Characteristics	Healthcare Needs			Barrier to Healthcare (vs. Met Healthcare Needs)								
				Financial Difficulty			Time Constraint			Lack of Caring and Support		
	Coeff	(SE)	*p*-Value	Coeff	(SE)	*p*-Value	Coeff	(SE)	*p*-Value	Coeff	(SE)	*p*-Value
Sociodemographics												
Man (R: woman)	−0.395	(0.043)	<0.001	0.018	(0.056)	0.754	−0.064	(0.033)	0.056	−0.033	(0.051)	0.521
Age, 0	0.148	(0.014)	<0.001	−0.035	(0.017)	0.034	−0.085	(0.023)	<0.001	0.060	(0.017)	0.001
Age-squared, 000	-	-	-	−0.179	(0.077)	0.020	−0.192	(0.073)	0.008	0.062	(0.072)	0.387
Man*age, 0 (R: woman*Age)	-	-	-	0.037	(0.022)	0.084	−0.039	(0.019)	0.037	0.001	(0.019)	0.976
Man*age-squared, 000 (R: woman*Age-squared)	0.388	(0.073)	<0.001	−0.188	(0.116)	0.105	−0.257	(0.096)	0.008	−0.261	(0.105)	0.013
Woman*age-squared, 000	0.114	(0.074)	0.122	-	-	-	-	-	-	-	-	-
Unmarried (R: married)	−0.294	(0.034)	<0.001	0.279	(0.043)	<0.001	0.039	(0.028)	0.164	0.087	(0.044)	0.049
Seoul metropolitan area (R: The other areas)	−0.254	(0.026)	<0.001	0.387	(0.036)	<0.001	0.050	(0.021)	0.019	0.173	(0.039)	<0.001
College or higher (R: Less than college)	0.064	(0.031)	0.040	−0.225	(0.053)	<0.001	−0.051	(0.027)	0.058	0.037	(0.043)	0.390
Occupation (R: no job)												
Blue-collar job	0.053	(0.036)	0.137	0.042	(0.044)	0.339	0.477	(0.120)	<0.001	−0.040	(0.039)	0.308
White-collar job	0.076	(0.041)	0.059	−0.017	(0.068)	0.805	0.478	(0.123)	<0.001	−0.251	(0.063)	<0.001
Household income (R: lowest quintile)												
Medium	0.055	(0.045)	0.214	−0.579	(0.042)	<0.001	−0.003	(0.032)	0.921	−0.138	(0.044)	0.002
Highest quintile	0.057	(0.051)	0.263	−1.178	(0.073)	<0.001	−0.077	(0.038)	0.043	−0.164	(0.056)	0.004
Medical Care Aid (R: National Health Insurance)	0.299	(0.106)	0.005	0.142	(0.075)	0.058	−0.156	(0.089)	0.079	0.203	(0.083)	0.015
Private health insurance, yes (R: No)	0.083	(0.035)	0.017	−0.152	(0.045)	0.001	0.027	(0.028)	0.327	−0.002	(0.041)	0.958
Physical and health conditions												
Functional limitation, yes (R: no)	0.437	(0.112)	<0.001	0.389	(0.053)	<0.001	0.021	(0.050)	0.672	0.655	(0.107)	<0.001
Current smoker, yes (R: no)	−0.122	(0.033)	<0.001	0.234	(0.050)	<0.001	0.090	(0.035)	0.011	0.164	(0.054)	0.002
Alcohol consumer, yes (R: no)	-	-	-	0.029	(0.037)	0.433	0.116	(0.037)	0.002	0.078	(0.035)	0.028
An active routine of physical exercise activity, yes (R: no)	0.021	(0.026)	0.425	-	-	-	-	-	-	-	-	-
Obese, yes (R: no)	−0.036	(0.029)	0.212	-	-	-	-	-	-	-	-	-
Poor self-assessed health, yes (R: no)	0.332	(0.062)	<0.001	0.634	(0.040)	<0.001	0.312	(0.074)	<0.001	0.453	(0.070)	<0.001
Chronic disease, yes (R: no)	0.418	(0.032)	<0.001	−0.028	(0.048)	0.565	0.028	(0.023)	0.229	−0.156	(0.049)	0.001
Constant	1.930	(0.063)	<0.001	−2.118	(0.074)	<0.001	−1.724	(0.430)	<0.001	−2.327	(0.348)	<0.001

Notes: Coeff denotes the estimates of coefficients. SE denotes robust estimates of standard errors. R denotes the reference category. The estimates of coefficients and their standard errors related to the age variable were displayed after the estimates were multiplied by 10 or 1000 because their magnitude was very small. The second, third, and fourth columns display the results compared to the associations between peoples’ characteristics and the people’s experience of meeting healthcare needs because the latter was chosen as the reference alternative of the quadchotomous outcome variable. All values were estimated using a complex sampling design. All characteristics were considered to be time-dependent. Source: The Korea Health Panel survey data (2014–2018).

## Data Availability

Data are from the Korea Health Panel survey, which is available to the scientific community with a signed data access agreement from the Korea Institute for Health and Social Affairs and the National Health Insurance Service database (https://www.khp.re.kr:444/eng/main.do, accessed on 20 May 2020).

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
