# Peer review of "Changes in Barriers That Cause Unmet Healthcare Needs in the Life Cycle of Adulthood and Their Policy Implications: A Need-Selection Model Analysis of the Korea Health Panel Survey Data"

_healthcare, 2022, doi:10.3390/healthcare10112243_

Round 1

Reviewer 1 Report

This is a very comprehensive analysis of panel data on an important healthcare issue pertaining to the unmet healthcare needs of Koreans.  The findings are logically presented.  The statistical analysis of panel data reveals important and relevant findings regarding the impact of non-financial barriers to healthcare service utilization. There are several issues on theoretical and methodological contributions of this panel study are noted as follows:

1. Theoretical Framework:  It is a well-known fact that social and personal determinants of health service use can be more systematical identified to guide the selection of social and personal factors influencing the unmet needs. For example, the behavioral system model of healthcare use developed by Ron Andersen and John Newman could be cited and used to explain how societal and individual factors could directly or indirectly influence healthcare use and outcomes.

2. Methodological Issues:  Table 1 shows that the cross-sectional and panel analysis of data.  However, it is unclear if both time-varying and time- constant predictors are used.  Because multiple waves of panel data are available, it is imperative to show the trajectories or patterns of healthcare use and outcomes.  Paul Allison's work was cited, but no detailed on the trajectories are presented.

3. Definition of Unmet Healthcare Needs:  The concept of "unmet needs" should be defined in the first section of the introduction.

4.  Limitations and Future Research:  The cross-sectional analysis is nicely done, but the panel data analysis could be strengthened if the latent growth curve model is used.

Overall, I think that this is a fine paper.  However, more insight could be gained if the trajectories of unmet needs and other outcomes are being considered.

Reviewer 2 Report

Thank you very much for sending me this manuscript. This is an exceptionally well-written article. I have a few minor comments.

1. I really liked that the author discussed some policy issues at the end of the paper. These issues to some extent point to the importance of understanding the dynamics of health care within health facilities. I think the paper can improve if the author can suggest some innovative ways for future research to further examine these issues.

2. Healthcare needs seem to be stratified by a number of demographic factors such as age and gender. It may be important to further discuss the ways for relevant policies to address these different needs. 

3. Please discuss the attrition rate of the survey if necessary. I wonder the proportion of people who were kept in the survey throughout multiple waves. Was there any way to adjust for this?

Round 2

Reviewer 1 Report

It is a fine revision. The limitation section has appropriately documented the relevant elements of future research.  I hereby recommend that the paper be accepted for publication.